# Deep Learning Framework with Multi-Head Dilated Encoders for Enhanced Segmentation of Cervical Cancer on Multiparametric Magnetic Resonance Imaging

**DOI:** 10.3390/diagnostics13213381

**Published:** 2023-11-03

**Authors:** Reza Kalantar, Sebastian Curcean, Jessica M. Winfield, Gigin Lin, Christina Messiou, Matthew D. Blackledge, Dow-Mu Koh

**Affiliations:** 1Division of Radiotherapy and Imaging, The Institute of Cancer Research, London SW7 3RP, UK; reza.kalantar@icr.ac.uk (R.K.); jessica.winfield@icr.ac.uk (J.M.W.); christina.messiou@rmh.nhs.uk (C.M.); mu.koh@icr.ac.uk (D.-M.K.); 2Department of Radiology, The Royal Marsden Hospital, London SW3 6JJ, UK; 3Department of Radiation Oncology, Iuliu Hatieganu University of Medicine and Pharmacy, 400347 Cluj-Napoca, Romania; sebastian.curcean@iocn.ro; 4Department of Medical Imaging and Intervention, Chang Gung Memorial Hospital at Linkou, Chang Gung University, Guishan, Taoyuan 333, Taiwan; giginlin@cgmh.org.tw

**Keywords:** tumor segmentation, multiparametric MRI, cervical cancer, deep learning, dilated convolution, radiology, radiation oncology

## Abstract

T_2_-weighted magnetic resonance imaging (MRI) and diffusion-weighted imaging (DWI) are essential components of cervical cancer diagnosis. However, combining these channels for the training of deep learning models is challenging due to image misalignment. Here, we propose a novel multi-head framework that uses dilated convolutions and shared residual connections for the separate encoding of multiparametric MRI images. We employ a residual U-Net model as a baseline, and perform a series of architectural experiments to evaluate the tumor segmentation performance based on multiparametric input channels and different feature encoding configurations. All experiments were performed on a cohort of 207 patients with locally advanced cervical cancer. Our proposed multi-head model using separate dilated encoding for T_2_W MRI and combined b1000 DWI and apparent diffusion coefficient (ADC) maps achieved the best median Dice similarity coefficient (DSC) score, 0.823 (confidence interval (CI), 0.595–0.797), outperforming the conventional multi-channel model, DSC 0.788 (95% CI, 0.568–0.776), although the difference was not statistically significant (*p* > 0.05). We investigated channel sensitivity using 3D GRAD-CAM and channel dropout, and highlighted the critical importance of T_2_W and ADC channels for accurate tumor segmentation. However, our results showed that b1000 DWI had a minor impact on the overall segmentation performance. We demonstrated that the use of separate dilated feature extractors and independent contextual learning improved the model’s ability to reduce the boundary effects and distortion of DWI, leading to improved segmentation performance. Our findings could have significant implications for the development of robust and generalizable models that can extend to other multi-modal segmentation applications.

## 1. Introduction

Cervical cancer is the fourth most common cancer in women worldwide [1]. In the epidemiological data of 2020, a total of 604,127 cervical cancer diagnoses were reported globally [2]. Despite the evolution in diagnostic and therapeutic modalities, projections for 2023 indicate a persistent challenge, with an anticipated 13,960 cases of invasive cervical cancer in the United States alone. This forecast also predicts a consequent 4310 mortality events [3]. Such statistics emphasize the imperative of advancements in precise diagnostic measures, including state-of-the-art imaging and segmentation methodologies.

Multiparametric magnetic resonance imaging (mpMRI) is a crucial tool in the diagnosis and management of gynecological malignancies, including cervical cancer. It provides detailed anatomical and functional information, which is applied for disease staging, treatment planning, response monitoring, and surveillance for disease recurrence [4,5,6]. An important aspect of many mpMRI protocols is diffusion-weighted imaging (DWI), which enhances the contrast and visualization of cellular tumors. DWI is sensitive to the rate of diffusion of water molecules in vivo, and offers insights into the intricate tissue microenvironment. Specifically, it has been shown to relate to cellularity, i.e., the density of cells within a tumor, and is also sensitive to microstructural alterations, such as necrosis and changes in the extracellular matrix, which can influence water diffusion patterns [7]. The rate of diffusion can be quantified at each spatial location through the estimation of maps of the apparent diffusion coefficient (ADC). ADC measurements offer the potential to used as a surrogate biomarker of tumor grade [8,9], and have shown promising results in identifying early treatment responses, making them desirable for monitoring therapeutic outcomes in cervical cancer [10,11]. In conventional anatomical T_2_-weighted (T_2_W) MRI, primary and metastatic tumors exhibit intermediate to high signal intensities. This is used to identify cervical abnormalities, as well as for disease staging and directing MRI-guided interventions [12,13].

Image segmentation in medical imaging involves the demarcation of regions of interest (ROIs) into semantically and clinically meaningful areas. Automating disease detection and delineation in medical images is a critical task, primarily because it aids in extracting valuable biomarkers from images, which enhances clinical decision-making. This process is currently impeded by the requirement for extensive annotated datasets, leading to a high dependency on clinicians and potential inconsistencies due to human contouring variations. Automated tumor segmentation on mpMRI therefore holds great significance, not only for reducing the burden on clinicians but also for its potential in improving accuracy and consistency. Furthermore, in contexts such as treatment planning, where manual delineation may not be feasible due to time constraints, these automated methods become particularly indispensable. Therefore, the development of fully automatic tumor segmentation techniques is a crucial step towards achieving more efficient and reliable clinical processes. Beyond operational benefits, this could also translate to profound patient-centric advantages, including shorter and more comfortable treatments, as well as more accurate and consistent treatment delivery. Such advancements could potentially mitigate collateral toxicity and optimize treatment outcomes.

Among these segmentation techniques, U-Net [14] stands out as a prevalent architecture for image segmentation. Numerous derivatives of U-Net have been developed for various medical segmentation tasks, with a particular emphasis on neuroimaging. These include the multi-scale densely connected U-Net (MDU-Net) [15], U-Net with interconnected skip connection pathways (U-Net++) [16], U-Net with residual extended ski connection and wide context modules (BU-Net) [17], U-Net with a feature enhancer block (BrainSeg-Net) [18], multi-scale recurrent residual U-Net with dense skip connections (R2U++) [19], the redesigned U-Net with full-Scale feature fusion and ghost modules (half-U-Net) [20], U-Net augmented with multi-scale feature fusion (SegR-Net) [21], U-shaped architecture with residual spatial pyramid pooling modules and attention gates (RAAGR2-Net) [22], and the pyramid dual-decoder attention U-Net (PDAtt-UNet) [23].

In parallel, architectures based on the vision transformer model [24] have shown promise for the accurate segmentation of medical images. For instance, Wang et al. [25] developed the mixed transformer U-Net for multi-organ segmentation in computed tomography (CT) images. Furthermore, Huang et al. [26] proposed a transformer-based generative adversarial network for multimodal brain tumor segmentation in MRI.

Despite these advancements, a significant challenge persists in accurately segmenting small-scale tumors in mpMRI images, particularly when there is spatial misalignment between the input channels of the models. This issue is especially prominent in pelvic images, where internal tissues are prone to spatial deformation during scanning sessions. Such deformations could impede the segmentation performance of networks that rely on conventional multi-channel inputs.

While current segmentation architectures demonstrate promising versatility for complex medical segmentation applications, there remains a scarcity of research on cancer tumor segmentation utilizing multiparametric MRI, especially in the context of cervical cancer. The segmentation of cervical cancer presents several challenges. These include the complex anatomy of the pelvis with closely situated organs, the appearance of inflammatory processes that can mimic or obscure tumor regions, and the potential presence of metastases that might be intermingled with the primary tumor or situated nearby. Moreover, the inherent variability of tumor appearances, challenges arising from low image resolutions or noisy DWI images, and the intricate distinctions between the primary cervical mass and adjacent lymph nodes further complicate the segmentation process.

Several studies have explored the DL-based segmentation of cervical cancer in MRI [27,28,29,30,31]. Specifically, some studies have combined semantic knowledge between T_2_W MRI and DWI/ADC in this context. Among related studies, Lin et al. [32] developed a U-Net model for segmenting cervical cancer in DWI and ADC images, and found that multi-channel input (b0, b1000, and ADC) produced the best segmentation performance. However, this study focused on two-dimensional (2D) images and did not incorporate multimodal MRI (e.g., MRI images derived from different sequences). Kano et al. [33] combined 2D and three-dimensional (3D) U-Net models for cervical tumor segmentation on DWI images using an ensembling approach. Yoganathan et al. [34] segmented primary tumors along with organs-at-risk (OARs) on T_2_W MRI, reporting that integrating segmentations from 2.5D training in axial, coronal, and sagittal planes improved segmentation performance compared with previous 2D models. However, this study was limited to 39 patients and single-channel inputs. Wang et al. [29] proposed a 3D CNN model for cervical tumor segmentation on multimodal MRI, while Hodneland et al. [35] utilized a U-Net with residual connections, employing T_2_W MRI, b1000 DWI, and ADC as input channels. However, none of these studies examined the impact of spatial mismatch between multimodal MRI inputs, resulting from distortion in echo-planar imaging (EPI) and soft-tissue deformations between scans [36,37], on cervical tumor segmentation outcome.

The aim of this study was to develop a novel 3D framework based on the U-Net architecture that included multi-head dilated residual encoding blocks for combined fine-grained and contextual feature aggregation and training on anisotropic sub-volumes of images, enhancing the segmentation of locally advanced cervical tumors on multiparametric MRI. To our best knowledge, no previous studies have incorporated and investigated this strategy for the automated segmentation of pelvic malignancies.

## 2. Materials and Methods

### 2.1. Patient Populations and Imaging Parameters

In this study, we utilized a retrospective cohort of 207 patients diagnosed with cervical cancer. The histopathology types of enrolled patients included squamous cell, adenosquamous carcinomas (either non-keratinizing or keratinizing), and all were HPV-associated, according to the World Health Organization (WHO) classification of female genital tumors introduced in the 4th edition (2014) [38]. The categorization of locally advanced cervical cancer was based on the 2018 guidelines set by the International Federation of Gynaecology and Obstetrics (FIGO) [39]. These patients underwent pelvic T_2_W MRI, and DWI on a 3T MAGNETOM TrioTim MRI scanner (Siemens Healthcare, Erlangen, Germany) was selected. The ground-truth tumor contours were defined by a clinician with 3 years of relevant experience on T_2_W MRI images, with the DWI data available for all patients. The acquisition parameters for this dataset are shown in Table 1. The ADC maps were calculated from DWI images with varying diffusion-weighting magnitude (b-value) (Equation (Equation 1)), using a mono-exponential fit for two b-values (b0, b1000) (Equation (Equation 2)) and least-square exponential fit for multiple b-values (b200, b600, b1000) (Equation (Equation 3)).
(1)$$S_{b_{i}}=S_{b_{0}}\cdot e^{\left(-b_{i}D\right)}$$
(2)$$D=-\frac{1}{b_{i}}\ln\left(\frac{S_{b_{i}}}{S_{b_{0}}}\right)$$
(3)$$D=-\frac{N\sum_{i=1}^{N}b_{i}\ln\left(S_{b_{i}}\right)-\sum_{i=1}^{N}b_{i}\sum_{i=1}^{N}ln\left(S_{b_{i}}\right)}{N\sum_{i=1}^{N}b_{i}^{2}-{\left(\sum_{i=1}^{N}b_{i}\right)}^{2}}$$
where $b_{i}$ is the b-value, $S_{b_{0}}$ represents the signal intensity with no diffusion weighting (b=0), $S_{b_{i}}$ is the signal intensity at $b_{i}$, *N* is the number of b-values and *D* denotes the ADC value.

### 2.2. Network Topology and Architectural Experiments

In this study, we employed a residual U-Net model, a modified version of the conventional U-Net [14], as the benchmark for our segmentation framework. The model architecture was based on an encoder–decoder structure with symmetrical skip connections at each level (Figure 1). Each encoding level consisted of two residual blocks with 3D convolutional layers followed by instance normalization and a parametric ReLU activation layer (PReLU). Each upsampling block included two residual blocks with 3D strided transposed convolutions and skip connection concatenation layers. The architecture included four downsampling steps with kernel filters of 32, 64, 128, and 256 and a feature map depth of 512 in the bottleneck. Downsampling operation was performed in the first convolutional layer in the block (stride = 2). The model was trained on (i) T_2_W MRI-only; (ii) T_2_W and ADC; and (iii) T_2_W, ADC, and b1000 DWI training data, with input channels set to 1, 2, and 3, respectively.

To address the boundary effects of DWI in multi-channel training, as the result of distortions from EPI compared with morphological imaging, a series of architectural experiments were performed. In the first experiment, we replaced the first encoder block in the baseline multi-channel model with three encoding heads. The first head utilized a non-dilated 3D convolutional operation on T_2_W MRI inputs only. The other two heads included 3D convolution operations with dilations of 2 and 4 respectively, accepting three input channels: T_2_W, b1000 DWI, and ADC. These heads were connected by residual weight sharing and feature activation summation nodes (multi-head model 1).

In the second multi-head model, we employed the separate encoding of T_2_W MRI, and combined b1000 DWI and ADC images while still maintaining weight sharing. The dilated heads in this model only included the b1000 DWI and ADC channels. This was performed to investigate the impact of these channels on the overall contextual learning (multi-head model 2).

Finally, in the third experiment, we employed a multi-head encoding strategy with no weight sharing between the T_2_W and b1000 DWI/ADC heads. Similar to the approach used for the second model, we applied dilated convolutions exclusively to the b1000 DWI and ADC images to facilitate independent contextual learning and reduce boundary-based attention. However, this architecture included an additional concatenation layer followed by a convolutional layer for dimensionality reduction and feature summation with the T_2_W head (multi-head model 3).

### 2.3. Image Pre-Processing and Implementation Details

Prior to training, all mpMRI images were resampled to an in-plane resolution of 0.6 × 0.6 mm^2^, which was the most common T_2_W MRI resolution in the dataset (approximately 85%), and a slice thickness of 4 mm. Bilinear interpolation was applied for the resampling of T_2_W and b1000 DWI images, as it allows for smooth intensity transitions, preserving the details within the images. On the other hand, ADC images and contours were resampled using the nearest-neighbor interpolation method. This method was selected due to its ability to retain the original discrete values of the images, which is crucial for ADC maps that are quantitative in nature, and for contours as they represent categorical labels or boundaries in segmentation tasks. Each channel was independently normalized to a mean of zero and unit variance. Finally, the dataset was randomly split into 157, 25, and 25 patients for training, validation, and testing, respectively.

During training, sub-volumes of size 256 × 256 × 16 voxels were extracted stochastically from the training data, ensuring that each patch contained at least one annotated tumor slice. Random data augmentation operations, such as intensity shifting, scaling, and cropping, were applied to improve network generalizability. The models were trained for 100,000 iterations using the Dice loss function, which outperformed the combined Dice and cross-entropy and Tversky [40] losses during the initial experiments on input channels. The Adam optimizer with an initial learning rate of 1 $\times \;10^{-4}$ and weight decay of 1 $\times \;10^{-5}$ was used, and a cosine annealing learning rate scheduler was employed after each epoch. Validation was performed after each epoch, based on the Dice scores of whole image volumes, and the best-performing weights were saved. Volumetric segmentations were generated using a sliding window algorithm with a 75% overlap between adjacent patches. The models were evaluated using the Dice similarity coefficient (DSC), 95th percentile Hausdorff distance (HD), mean surface distance (MSD), and percentage relative volume similarity metrics [41]. PyTorch and Monai [42] DL libraries were used for all implementations.

### 2.4. Channel Sensitivity Analysis and Visualization

To assess the significance of individual channels in our models, we conducted a channel sensitivity analysis using sequential channel dropout. By setting each channel to zero one at a time, we compared the segmentation results obtained from the baseline multi-channel and proposed multi-head models to those achieved with no channel dropout. We employed the same quantitative metrics utilized in our previous analyses to perform a comprehensive comparison between the proposed architectures. To delve deeper into the channel-wise significance within our models, we incorporated a 3D adaptation of Gradient-weighted Class Activation Mapping (GRAD-CAM) [43]. GRAD-CAM serves as a method to visually interpret the relevance of particular regions within an input image in relation to the final prediction of a DL model. It does so by leveraging the gradient information flowing into the last convolutional layer of the model to generate a coarse localization map, thereby highlighting regions of interest.

In the context of our research, we tailored this technique to cater to the 3D data modality. The produced saliency maps from the penultimate layers of the investigated models explicitly emphasized the areas of highest importance for the segmentation tasks at hand. Specifically, when this technique was applied to the center-cropped patches sourced from the test image volumes, the resulting visualizations not only detailed the salient features but also provided insight into the channel-wise contributions. Such intricate visual insights are paramount in decoding the rationale behind the model’s predictions and further fine-tuning its segmentation capabilities.

## 3. Results

The baseline multi-channel model was evaluated on various input channels and loss functions. Multi-channel input trained with Dice loss achieved the best mean segmentation performance in terms of DSC across all test cases, with consistent results across input channels. The average DSC values for multi-channel input (T_2_W, b1000 DWI, ADC) using Dice, combined Dice and cross-entropy, and Tversky losses were 0.672, 0.661, and 0.664, respectively (Figure 2).

The comparison of segmentations obtained from the baseline multi-channel and the proposed multi-head models revealed that multi-head model 3 exhibited superior performance across the quantitative metrics analyzed (Figure 3a–d). Specifically, the median DSC values for the multi-channel model, multi-head model 1, multi-head model 2, and multi-head model 3 were 0.788 (confidence interval (CI), 0.568–0.776), 0.805 (CI, 0.538–0.769), 0.796 (CI, 0.537–0.776), and 0.823 (CI, 0.595–0.797), respectively (Figure 3a). However, the differences in performance between the proposed multi-head models were not significantly different from the multi-channel model (*p* > 0.05). On average, all models underestimated tumor volume compared with the contours delineated by the clinician, with median relative percentage volume differences for each model of −14.4%, −18.9%, −9.7%, and −12.0%, respectively (Figure 3d). Multi-head model 3 demonstrated the best quantitative scores compared with other experimental architectures; therefore, the segmentation contours predicted by this model were compared with those obtained from the baseline multi-channel model (Figure 4).

A channel sensitivity analysis was performed to compare the performance metrics between the multi-channel and our proposed multi-head model. One finding was the significant dependence of the multi-head model on the ADC channel. Specifically, when the ADC channel was excluded from the input, the tumor segmentation performance of the multi-head model deteriorated markedly, as evidenced in Figure 5. The saliency maps generated from both models revealed that b1000 DWI images had a relatively minor impact on the overall tumor segmentation performance for larger tumor volumes. In contrast, the T_2_W and ADC images were more crucial, as visualized in Figure 6, test cases 1 and 2. However, for smaller and more challenging tumor masses, the absence of ADC channels had a more pronounced adverse effect on the final outcome (see Figure 6, test cases 3–5).

## 4. Discussion

DWI is a critical functional imaging technique for the detection and localization of tumors. This is particularly evident in ADC maps, where regions depicting impeded water molecule diffusion are indicative of increased cell density, often signifying a more aggressive disease [44]. However, using DWI and ADC maps in conjunction with T_2_W MRI sequences presents several challenges, such as voxel misalignment due to distortion and soft-tissue deformations between scans, variations in tumor delineations between the two modalities and the absence of standardized protocols for mpMRI. In this study, we proposed a novel multi-head framework that utilizes both b1000 DWI (including ADC) and T_2_W MRI for cervical cancer segmentation—a process well-suited to potential biomarker extraction. Our framework includes separate encoding heads that extract contextual information about tumors using dilated (or atrous) convolutions and shared residual connections. We have demonstrated that our technique provides a more robust and boundary-aware segmentation of cervical tumors when compared with the baseline multi-channel architecture that is commonly used in previous studies [32,35]. Our findings using the multi-channel training approach are comparable to previous reports on the segmentation of cervical tumors in multiparametric MRI [29,32,35]. However, it is challenging to make direct comparisons of our results due to the lack of public databases and resources for cervical cancer segmentation in MRI [45] and the use of different datasets.

Dilated convolutions are a crucial component of several successful segmentation architectures, including DeepLabv3+ [46], DeepLab [47], residual enhanced U-Net and atrous convolution (AtResU-Net) [48], and 3D deeply supervised fully convolutional network with concatenated atrous convolution (3D DSA-FCN) [49]. While most of these techniques utilize dilated convolutions throughout their architectures’ encoding steps, we propose a multi-head framework that uses these operations for separate contextual and representational learning in b1000 DWI and ADC images for only the first block of the encoder. This approach ensures that the training parameters are not increased drastically compared with baseline multi-channel architectures, and the model is easier to interpret with conventional methods for future explainability studies. Moreover, lighter networks are better suited to online MR-guided treatments, where segmentation and planning are performed live on the scan taken on the day before radiation treatment [50], and speed is of the essence. In this study, we trained our models using anisotropic sub-volumes to maintain a greater focus on the plane of acquisition (axial) for 2D MRI. Our presented methodology, while primarily focused on its current application, possesses a versatile architecture that can be adapted and extrapolated for broader utilities. For example, it can be extended to 3D MRI scans, which are more commonly used for radiation therapy.

Channel sensitivity and saliency mapping of our experimental model indicated that our algorithm was more sensitive to ADC maps, which potentially makes it more robust to changes in acquisition protocol in MRI scanners. This approach could also serve as a strategy for more generalizable and cross-disease detection models [51]. However, this dependence on b1000 DWI and ADC images, as demonstrated in this study, may result in underestimation of the predicted tumor volume for malignancies with heterogeneous tumor mass diffusion. Although our findings indicate a stronger dependence on ADC input channels, further investigations are required for a more comprehensive understanding of tumor segmentation outcome using MRI and DWI images from multiple centres. Moreover, the subjectivity associated with inter-operator variability presents another drawback, with more reliable segmentations only attainable through the use of consensus contours. The decision to include specific areas within the ground-truth contour of the tumor is a discretionary choice made by the clinical expert annotating the images—a decision that relies heavily on their professional training and experience. It is important to note that this limitation may bias the findings from this study. Hence, future studies should aim to employ consensus ground-truth contours and evaluate the segmentation outcome through a number of expert human reader assessments to ensure the accuracy and reliability of the results for clinical decision-making.

## 5. Conclusions

Our proposed multi-head framework that combines b1000 DWI, ADC, and T_2_W MRI for cervical cancer segmentation has demonstrated improved accuracy and robustness compared to conventional multi-channel architectures. The use of dilated convolutions in only the first block of the encoder improves contextual learning with no significant parameter increase compared with conventional U-Net models. However, both the dependence on b1000 DWI and ADC channels and inter-operator variability are potential limitations that need to be addressed in future studies. Potential solutions may include architectural improvements or the use of consensus ground-truth contours and expert human reader assessments. Overall, our approach could offer a more accurate solution for monitoring disease progression and treatment response. This has the potential to enhance clinical decision-making in the diagnosis and treatment of patients with cervical cancer, as well as other pelvic malignancies.

## Figures and Tables

**Figure 1 diagnostics-13-03381-f001:**
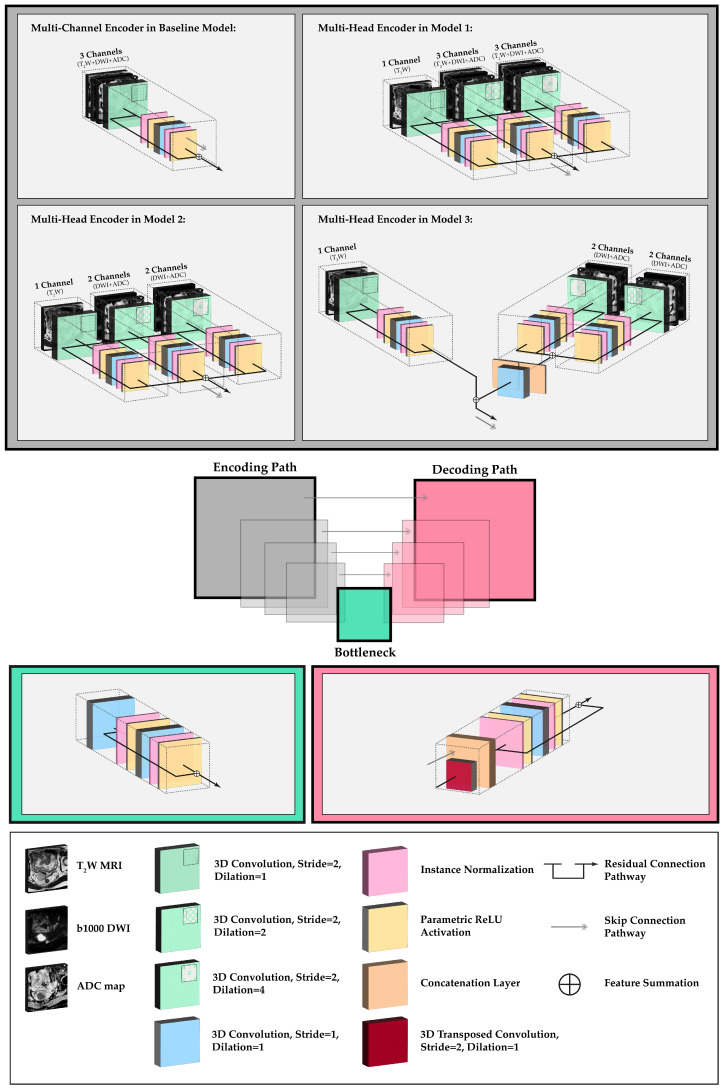
The network topology and the multi-head segmentation training experiments. The operational blocks incorporate residual connections to facilitate the flow of information between different layers. The multi-head models use various encoding and weight-sharing configurations for T_2_W MRI, b1000 DWI and ADC maps using multiple heads with dilated convolution and connective residual operations.

**Figure 2 diagnostics-13-03381-f002:**
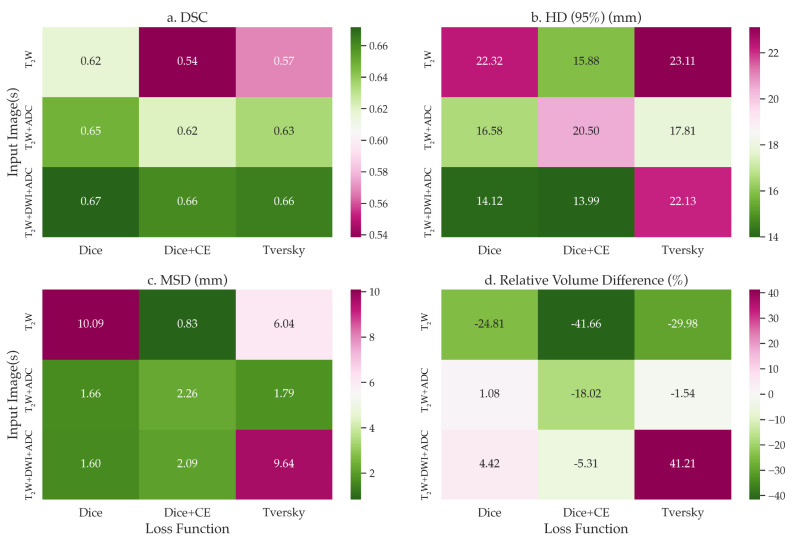
Heatmaps of mean (**a**) Dice similarity coefficient (DSC), (**b**) 95th percentile Hausdorff distance (HD), (**c**) mean surface distance (MSD), and (**d**) relative percentage volume difference for the multi-channel residual U-Net model trained across different input images and loss functions from the test data. The white color within the volume difference colourmap indicates the optimal metrics. The model trained on all T_2_W, b1000 DWI, and ADC channels, and with Dice loss, achieved the best overall segmentation performance across all experiments.

**Figure 3 diagnostics-13-03381-f003:**
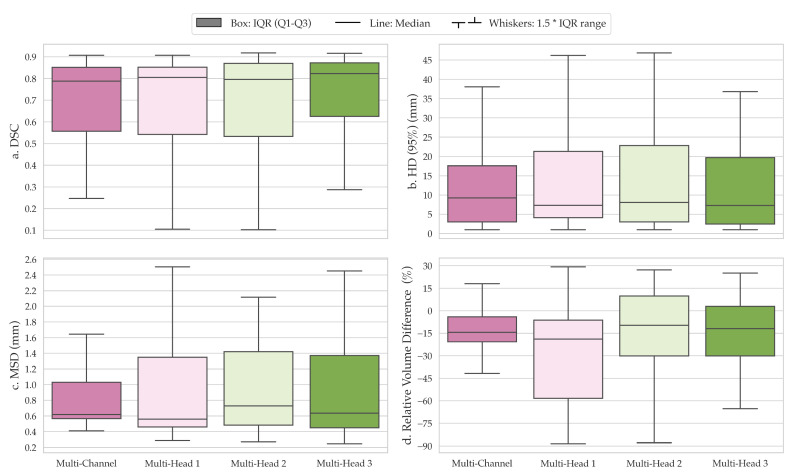
Comparison of segmentation performance based on (**a**) Dice similarity coefficient (DSC), (**b**) 95th percentile Hausdorff distance (HD), (**c**) mean surface distance (MSD), and (**d**) relative percentage volume difference between the baseline multi-channel model and three variations of the dilated multi-head model. Overall, the multi-head model with dilated convolutions and separate b1000 DWI/ADC feature aggregation (multi-head model 3) achieved the best performance among all models. IQR: interquartile range.

**Figure 4 diagnostics-13-03381-f004:**
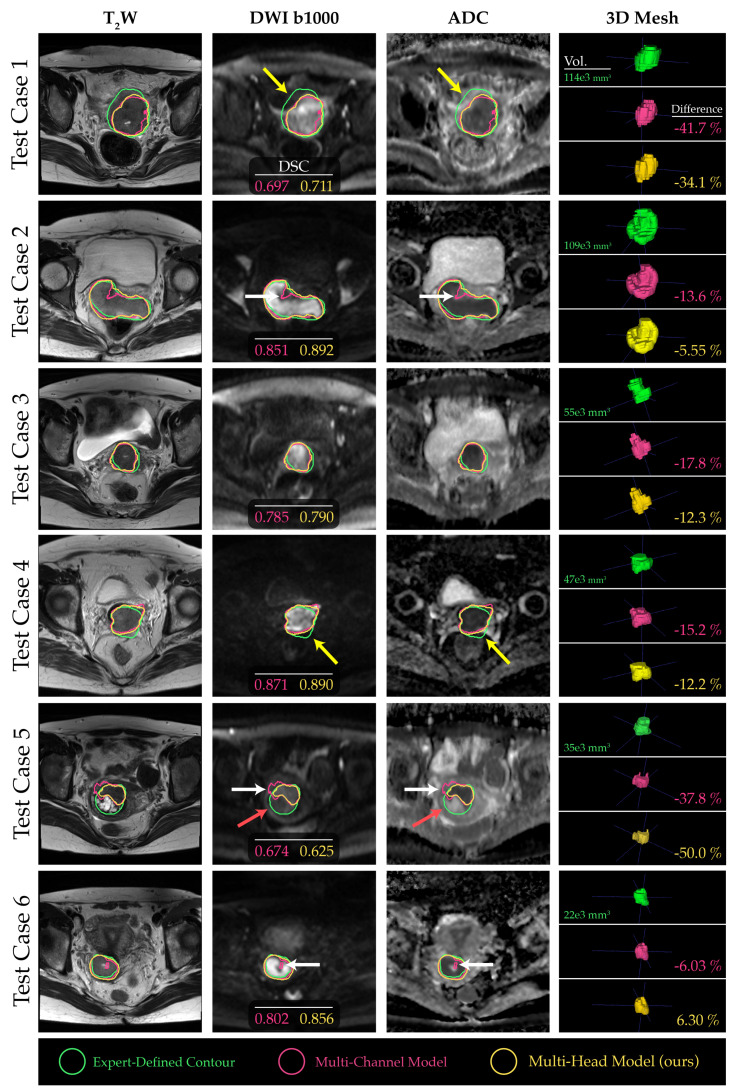
Comparative evaluation of the baseline multi-channel and proposed multi-head models for 6 example test cases, arranged in descending order of tumor size. The yellow arrows indicate regions where distortion in DWI and subjectivity in tumor location made contour propagation from T_2_W images challenging. Conversely, the white arrows highlight regions where the proposed model performed better by capturing boundaries and contextual information compared to the baseline multi-channel model. The red arrows highlight tumor regions within the ground-truth contours that did not exhibit strong impeded diffusion, thus remaining undetected by the DL models.

**Figure 5 diagnostics-13-03381-f005:**
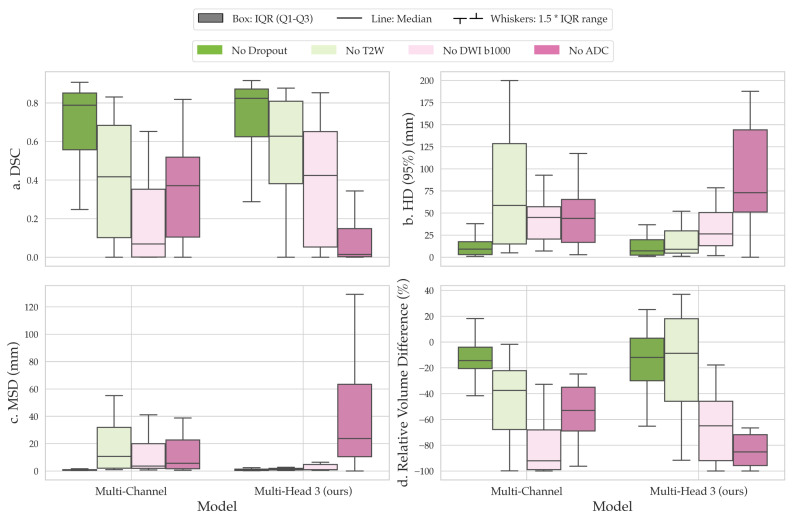
Comparison of the sensitivity of the multi-channel and proposed multi-head models to different input channels based on quantitative metrics, including the (**a**) Dice similarity coefficient (DSC), (**b**) 95th percentile Hausdorff distance (HD), (**c**) mean surface distance (MSD), and (**d**) relative percentage volume difference. Our channel dropout analysis revealed a strong dependence of the proposed multi-head model on the ADC channel, indicating its importance in achieving accurate segmentation performance. IQR: interquartile range.

**Figure 6 diagnostics-13-03381-f006:**
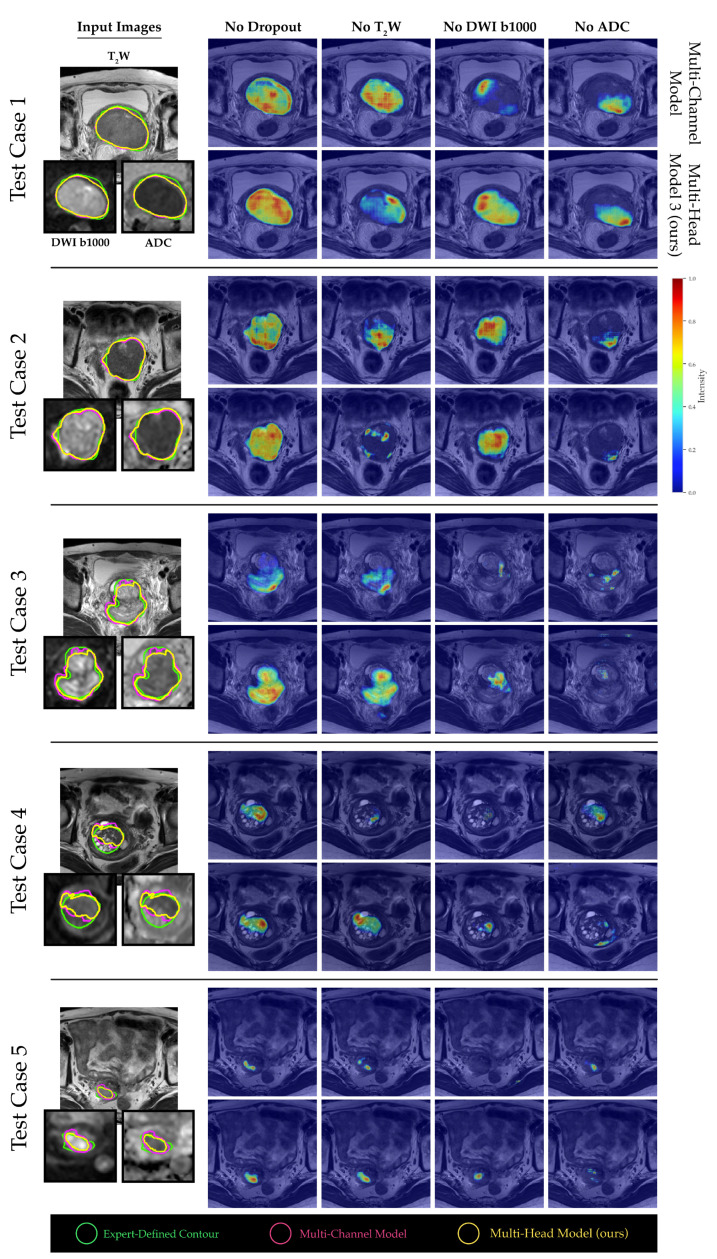
The normalized 3D GRAD-CAM saliency maps generated from the penultimate layer of the models, highlighting the most relevant regions for segmentation under different channel dropout conditions at test time. To improve visualization, the b1000 DWI and ADC images for smaller tumors were scaled.

**Table 1 diagnostics-13-03381-t001:** Imaging parameters of the cohort used in this study (*n* = 207). The values are presented as the range from minimum to maximum values, or are variables using different protocols within the dataset.

Parameter	T_2_W MRI	DWI
Manufacturer Name	Siemens Healthineers	Siemens Healthineers
Scanner Model	MAGNETOM Trio	MAGNETOM Trio
Magnetic Field Strength (T)	3	3
Sequence	Turbo Spin Echo (TSE)	Echo-Planar Imaging (EPI)
Slice Orientation	Axial	Axial
Echo Time (ms)	80–101	60–80
Repetition Time (ms)	3600–8060	3300–10,844
Acquired Matrix Size (read)	224–320	128–172
Reconstructed Matrix Size (read)	256–320	240–248
Reconstructed Pixel Size (mm${}^{2}$)	0.5 × 0.5–0.8 × 0.8	1.2 × 1.2–1.4 × 1.4
Slice Thickness (mm)	4.0–5.0	4.0–5.0
Flip Angle (°)	120–160	180
Phase Encoding Direction	Anterior–Posterior or Left–Right	Anterior–Posterior
Receiver Bandwidth (Hz/pixel)	190–200	1940–2441
b-values (s/mm${}^{2}$)	-	[0,1000] or [200,600,1000]

## Data Availability

The data for this study are unavailable due to privacy or ethical restrictions.

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
