# Peer review of "Deep Learning Framework with Multi-Head Dilated Encoders for Enhanced Segmentation of Cervical Cancer on Multiparametric Magnetic Resonance Imaging"

_diagnostics, 2023, doi:10.3390/diagnostics13213381_

Round 1
Reviewer 1 Report
Comments and Suggestions for Authors
I believe the manuscript is well but the manuscript is not represented well. I really want the manuscript to be published but after necessary improvements. Please address following concerns and again submit the manuscript so that it can be considered for publications,
@Please include references to recent U-Net variants in the manuscript to acknowledge the latest advancements in image segmentation techniques. Here are few of the recent U-Net Variants from literature (Especially from medical field) that should be in the manuscript,
@ SegR-Net: A deep learning framework with multi-scale feature fusion for robust retinal vessel segmentation
@ PDAtt-Unet: Pyramid dual-decoder attention Unet for Covid-19 infection segmentation from CT-scans
@ MDU-Net: multi-scale densely connected U-Net for biomedical image segmentation
@ Are there any potential biases or limitations in the automated segmentation approach that should be addressed?
@ Are there any unexpected findings or patterns in the data that should be explored further?
@ References in the manuscript are missing. Please check your latex file and fix it.
@ Please improve the introduction by adding more details from literature and some statistical details. You will get good details from UN reports. By carrying out literature you will get good details.
Comments on the Quality of English LanguageSome slight improvement is required.
Author Response
@Please include references to recent U-Net variants in the manuscript to acknowledge the latest advancements in image segmentation techniques. Here are few of the recent U-Net Variants from literature (Especially from medical field) that should be in the manuscript,
@ SegR-Net: A deep learning framework with multi-scale feature fusion for robust retinal vessel segmentation
@ PDAtt-Unet: Pyramid dual-decoder attention Unet for Covid-19 infection segmentation from CT-scans
@ MDU-Net: multi-scale densely connected U-Net for biomedical image segmentation
Thank you for your suggestion. This information was added to the fourth paragraph in the introduction section:
‘The advent of advanced imaging and high-performance technologies has led to a surge of interest in deep learning (DL) and convolutional neural network (CNN)-based approaches for medical image segmentation [14]. Among these, U-Net [15] stands out as a prevalent architecture for image segmentation. Its derivatives, including the multi-scale densely connected U-Net (MDU-Net) [16], U-Net with interconnected skip connection pathways (U-Net++) [17], U-Net augmented with multi-scale feature fusion (SegR-Net) [18], and the pyramid dual-decoder attention U-Net (PDAtt-UNet) [19], have been reported in the literature for diverse medical segmentation tasks.'
@ Are there any potential biases or limitations in the automated segmentation approach that should be addressed?
To further highlight the limitations of the proposed approach, more information was included in the discussion section:
‘This approach could also serve as a strategy for more generalizable and cross-disease detection models [43]. However, this dependence on b1000 DWI and ADC images, as demonstrated in this study, may result in underestimation of the predicted tumor volume for malignancies with heterogeneous tumor mass diffusion.’
‘Moreover, the subjectivity associated with inter-operator variability presents another drawback, with more reliable segmentations only attainable through the use of consensus contours. The decision to include specific areas within the ground truth contour of the tumor is a discretionary choice made by the clinical expert annotating the images, a decision that relies heavily on their professional training and experience. It is important to note that this limitation may potentially bias the findings from this study.’
@ Are there any unexpected findings or patterns in the data that should be explored further?
While this was briefly mentioned in the discussion section, the following statement was added to ensure validity of findings through more rigorous future research:
‘Despite our findings indicating the stronger dependence on ADC input channels, further investigations are required for more comprehensive understanding of tumor segmentation outcome using MRI and DWI images from multiple centres.’
@ References in the manuscript are missing. Please check your latex file and fix it.
Thank you for this. The Latex file for this was fixed and all references are now visible on the manuscript.
@ Please improve the introduction by adding more details from literature and some statistical details. You will get good details from UN reports. By carrying out literature you will get good details.
Thank you for your feedback. More information was added to the introduction section. The starting paragraph was added as follows:
‘Cervical cancer is the fourth most common cancer in women worldwide [1]. In the epidemiological data of 2020, a total of 604,127 cervical cancer diagnoses were reported globally [2]. Despite the evolution in diagnostic and therapeutic modalities, projections for 2023 indicate a persistent challenge, with an anticipated 13,960 cases of invasive cervical cancer in the United States alone. This forecast also predicts a consequent 4,310 mortality events [3]. Such statistics emphasize the imperative for advancements in precise diagnostic measures, including state-of-the-art imaging and segmentation methodologies.’
Furthermore, more information was added regarding the recent technologies inspired by the U-Net architecture that were shown to improve segmentation outcome on medical images.
Reviewer 2 Report
Comments and Suggestions for Authors
The article is interesting, has an elements of novelty and experiment is designed appropiately, but has no references.
I found some minor issues to correct/ explain:
Summary: I suggest put forward possible benefits of the research for patients.
Introduction:
What lies beneath 'diffusion' in DWI, other than a water diffusion and can be observed in tumours?
Please, describe in introduction what segmentation is, and how it can help.
What can be possible sources of failures in segmentation of cervical cancer MRI images? What about anatomy of pelvis, inflammatory processes and metastases?
Materials and methods:
Line 80: 'locally advanced cervical cancer' - all cancers were squamous cell carcinomas SCC)? non-keratinizing? HPV-associated? locally advanced - according which classification (name and year of classification)?
Figures 1, 2, 3 and 5 - enlarge the font, please.
Figures 3 and 5 - add legend of the chart (boxes and whiskers), please.
Author Response
The article is interesting, has an element of novelty and experiment is designed appropriately, but has no references.
Thank you for your feedback. The Latex file for references has been updated and added to the manuscript.
I found some minor issues to correct/ explain:
Summary: I suggest put forward possible benefits of the research for patients.
Thank you for your suggestion. More information was added to highlight the potential benefits to patients. More information was added within the paper, however more explicitly highlighted in the following statements:
Introduction - ‘Beyond operational benefits, this could also translate to profound patient-centric advantages, including shorter and less discomforting treatments, as well as more accurate and consistent treatment delivery. Such advancements could potentially mitigate collateral toxicity and optimize treatment outcomes.’
Conclusion – ‘Overall, our approach could offer a more accurate solution for monitoring disease progression and treatment response. This has the potential to enhance clinical decision-making in the diagnosis and treatment of patients with cervical cancer, as well as other pelvic malignancies.’
Introduction:
What lies beneath 'diffusion' in DWI, other than a water diffusion and can be observed in tumours?
For improved clarity, more information on DWI and its use for identifying the tumors were added in the introduction section:
‘DWI is sensitive to the rate of diffusion of water molecules in vivo, and offers insights into the intricate tissue microenvironment. Specifically, has been shown to relate to cellularity, i.e. the density of cells within a tumor, and is also sensitive to microstructural alterations, such as necrosis and changes in the extracellular matrix, which can influence water diffusion patterns [7].’
Please, describe in introduction what segmentation is, and how it can help.
Thank you for your suggestion. An introductory sentence to describe segmentation was added to paragraph 3 in the introduction section:
‘Image segmentation in medical imaging involves the demarcation of regions of interest (ROIs) into semantically and clinically meaningful areas. Automating disease detection and delineation on medical images is a critical task, primarily because it aids in extracting valuable biomarkers from images, which enhances clinical decision-making.’
Additionally, more information regarding the importance of segmentation is included within the same paragraph.
What can be possible sources of failures in segmentation of cervical cancer MRI images? What about anatomy of pelvis, inflammatory processes and metastases?
For more clarity, the following sentences were added to paragraph 4 of the introduction section to describe potential challenges of cervical cancer segmentation:
‘The segmentation of cervical cancer presents several challenges. These include the complex anatomy of the pelvis with closely situated organs, the appearance of inflammatory processes that can mimic or obscure tumor regions, and the potential presence of metastases which might be intermingled with the primary tumor or situated nearby. Moreover, the inherent variability of tumor appearances, challenges arising from low image resolutions or noisy DWI images, and the intricate distinctions between the primary cervical mass and adjacent lymph nodes further complicate the segmentation process.’
Materials and methods:
Line 80: 'locally advanced cervical cancer' - all cancers were squamous cell carcinomas SCC)? non-keratinizing? HPV-associated? locally advanced - according which classification (name and year of classification)?
To provide more accurate information, this sentence was changed to the following:
‘In this study, we utilized a retrospective cohort of 207 patients diagnosed with locally advanced cervical cancer. The histopathological types present in the patients were squamous cell and adenosquamous carcinomas, including both non-keratinizing and keratinizing variations. All of these types were HPV-associated, according to the 4th edition (2014) of the World Health Organization's (WHO) classification of female genital tumors [31]. The categorization of locally advanced cervical cancer was based on the 2018 guidelines set by the International Federation of Gynaecology and Obstetrics (FIGO) [32].’
Figures 1, 2, 3 and 5 - enlarge the font, please.
Figures 3 and 5 - add legend of the chart (boxes and whiskers), please.
Thank you for your suggestions. The fonts on the mentioned figures were increased (where possible) and legends added to Figures 3 and 5 for more clarity.
Round 2
Reviewer 1 Report
Comments and Suggestions for Authors
Thanks for addressing my concerns. The manuscript seems to be in highly improved form. My last concern is regarding the U-Net literature review paragraph you added. There is no flow of that paragraph. I want you to create a flow of the paragraph and at the end of the paragraph you should be mentioning the limitations by these models. For your ease I can tell you that these all models are not able to segment put the small scale tumor regions. You can add this stuff to finalize your manuscript. Moreover, you have picked your few references from year 2015, but you can get same reference from latest review. I suggest you can further incorporate following manuscripts to further strengthen up your view point,
- RAAGR2-Net: A brain tumor segmentation network using parallel processing of multiple spatial frames
- Mixed transformer u-net for medical image segmentation
- Brainseg-net: Brain tumor mr image segmentation via enhanced encoder–decoder network
- R2U++: a multiscale recurrent residual U-Net with dense skip connections for medical image segmentation
- Half-UNet: A simplified U-Net architecture for medical image segmentation
- Bu-net: Brain tumor segmentation using modified u-net architecture
Author Response
Thank you for your feedback. This paragraph was extended for more clarity as follows:
Among these, U-Net \cite{ronneberger2015u} stands out as a prevalent architecture for image segmentation. Numerous derivatives of U-Net have been developed for various medical segmentation tasks, with a particular emphasis on neuroimaging. These include the multi-scale densely connected U-Net (MDU-Net) \cite{zhang2018mdu}, U-Net with interconnected skip connection pathways (U-Net++) \cite{zhou2019unetpp}, U-Net with residual extended ski connection and wide context modules (BU-Net) \cite{rehman2020bu}, U-Net with a feature enhancer block (BrainSeg-Net) \cite{rehman2021brainseg}, multi-scale recurrent residual U-Net with dense skip connections (R2U++) \cite{mubashar2022r2u}, the redesigned U-Net with full-Scale feature fusion and ghost modules (half-U-Net) \cite{lu2022half}, U-Net augmented with multi-scale feature fusion (SegR-Net) \cite{ryu2023segr}, U-shaped architecture with residual spatial pyramid pooling modules and attention gates (RAAGR2-Net) \cite{rehman2023raagr2}, and the pyramid dual-decoder attention U-Net (PDAtt-UNet) \cite{bougourzi2023pdatt}.
In parallel, architectures based on the vision transformer model \cite{dosovitskiy2020image} have shown promise for accurate segmentation of medical images. For instance, Wang et al. \cite{wang2022mixed} developed the mixed transformer U-Net for multi-organ segmentation in computed tomography (CT) images. Furthermore, Huang et al. \cite{huang2022transformer} proposed a transformer-based generative adversarial network for multimodal brain tumor segmentation in MRI.
Despite these advancements, a significant challenge persists in accurately segmenting small-scale tumors in mpMRI images, particularly when there is spatial misalignment between the input channels of the models. This issue is especially prominent in pelvic images, where internal tissues are prone to spatial deformation during scanning sessions. Such deformations could impede the segmentation performance of networks that rely on conventional multi-channel inputs.
Reviewer 2 Report
Comments and Suggestions for Authors
The article is satisfactorily corrected but one - the legend of Figures 3 and 5 do not have boxes description (wiskers - SE/SD? box - min-max/Q1-3/other? line in the box - mean/median?) and group information is doubled (below x-axis and in legend).
Author Response
Thank you for your feedback. Figures 3 and 5 have been amended for more clarity.